# Preliminary Data from Six Years of Selective Anthelmintic Treatment on Five Horse Farms in France and Switzerland

**DOI:** 10.3390/ani10122395

**Published:** 2020-12-15

**Authors:** Liselore Roelfstra, Marion Quartier, Kurt Pfister

**Affiliations:** 1Laboratoire Animal Diagnostic, Beauregard 28, CH-2036 Corcelles-Cormondreche, Switzerland; liselore.roelfstra@animaldiagnostic.ch (L.R.); marion.quartier@animaldiagnostic.ch (M.Q.); 2Parasite Consulting GmbH, Wendschatzstrasse 8, CH-3006 Berne, Switzerland

**Keywords:** strongyle, selective anthelmintic treatment, epidemiology

## Abstract

**Simple Summary:**

Today, anthelmintic resistance (AR) of small strongyles (cyathostomins) against all presently available anthelmintics for equids poses on many horse farms worldwide huge problems with regard to an efficient and satisfactory parasite control. Therefore, alternative parasite control schemes are urgently needed. The so-called selective anthelmintic or targeted selective treatment (SAT) is one of the concepts considered to delay or even to overcome this challenging AR-situation. In the present field study, all 93 equids (90 horses, 3 ponies) from five horse riding farms in France and Switzerland were regularly sampled (spring and autumn) and the feces were analyzed for a period of six years. From a total of 757 fecal samples, only 263 (34.7%) had a fecal egg count ≥200 Eggs per Gram (EpG) (threshold) and consequently needed an anthelmintic treatment. A long-term reduction in the number of anthelmintic treatments can be expected on a herd and on the individual horse level, respectively, when comparing to a conventional (or strategic) twice per year treatment regime.

**Abstract:**

Anthelmintic resistance (AR) of small strongyle populations (cyathostomins) against products of the benzimidazole and tetrahydropyrimidine classes occurs now worldwide and there is an increasing number of reports also regarding macrocyclic lactones. Consequently, and in order to maintain an appropriate horse parasite control, alternative control schemes must be evaluated under field conditions. Here we present a six-year field study on the administration of the so-called selective or targeted selective anthelmintic treatment (SAT) concept. In this study on five horse farms in France and Switzerland, 757 fecal samples from 93 equids (90 horses, 3 ponies) have been taken twice a year (between early and late spring and between early and late autumn) from autumn 2014 to spring 2020 and processed by a McMaster technique. From a total of 757 samples, only 263 (34.7%) had a fecal egg count ≥200 EpG and needed an anthelmintic treatment. This small number of fecal samples ≥200 EpG demonstrates the considerable potential for a long-term reduction of the number of anthelmintic treatments and the anthelmintic pressure by using the SAT-programme.

## 1. Introduction

Anthelmintic resistance (AR) of small strongyle populations (cyathostomins) against products of the benzimidazole and tetrahydropyrimidine classes occurs now worldwide and had thus become a serious issue in horse parasite control [1,2,3,4,5]. Also, an increasing number of reports indicate that AR of cyathostomins to macrocyclic lactones is emerging [6,7,8,9] or, has recently been proven [10]. At present or in the near future, it seems that no new anthelmintic class aimed at horse helminths will become licensed. Consequently, and in order to maintain an appropriate and at the same time satisfactory control of horse parasites (cyathostomins but also other helminths), alternative control schemes are urgently needed and should be evaluated under field conditions.

One of the concepts considered to delay or even to overcome this challenging AR-situation is the so-called selective or targeted selective treatment procedure [11,12,13]. This latter treatment scheme aims, according to a compulsory preceding coprological diagnosis, to deworm only those individual animals exceeding a certain minimum level of the excreted strongyle eggs. According to various experiences [11,14,15], this number has been set to 200 eggs per gram of feces (EpG).

Following such a selective anthelmintic treatment (SAT) system allows (i) to take advantage of the concept of parasite refugia [16], i.e., leaving thereby a proportion of the worm population and susceptible worms, respectively, unexposed to anthelmintics on the pasture, and (ii) to considerably decrease the number of treatments and thus the quantity of anthelmintics administered within a herd [11,15,17].

Hence, the aim of this field study was to evaluate the long-term consistency and the outcome of a fully administered SAT-programme over a period of at least five years on several horse farms in Switzerland and France under field conditions.

## 2. Materials and Methods

### 2.1. Horse Farms and Animals

This retrospective investigation includes the fecal egg counting (FEC)—data from all 93 equids (90 horses, 3 ponies) from a total of five horse farms located in Switzerland and France who lived on these farms between autumn 2014 and spring 2020. All these farms perform the parasite control on their horses by using the Selective Anthelmintic Treatment (SAT) approach from autumn 2014 or early 2015. The horse farms were primarily riding stables with a paddock pasture system. From three farms (B, S, G) there was no regular collection of feces from the pasture, as the two other farms (C, N) collected at least once per week.

The age of the horses (mares and geldings) varied between 3 and 32 years.

All owners of the farms gave their positive consent to use the FEC-data of their horses for this study.

### 2.2. Fecal Sampling

All the animals were regularly and individually sampled twice a year, i.e., in spring and autumn by taking freshly deposited feces. Immediately after sampling, the feces were delivered to the laboratory for further processing.

### 2.3. Laboratory Analyses

Modified McMaster Method (strongyle fecal egg count: eggs per gram of feces = EpG).

From each sample, 4.0 g of cooled and thoroughly mixed up feces was processed by using a 1.2 saturated salt solution [18]. The sensitivity of the method is 25 EpG.

In order to detect other potential helminths, every fecal sample was simultaneously analyzed with a combined sedimentation/flotation technique [19].

## 3. Results

The presented data show that—even though the sampling periods from farm to farm were usually spread over several weeks—SAT control schemes were quite easily adaptable to such horse farm types and function accordingly with a high degree of consistency.

Over the six-year period, a total of 757 fecal samples from 93 equids (90 horses, 3 ponies) on five horse farms were collected and simultaneously analyzed by means of both, a McMaster technique and a sedimentation/flotation method, respectively.

The results are presented at the individual farm level in Table 1.

Out of all samples analyzed, only 34.7% had an EpG > 200. Thus, the number of diagnostically indicated anthelmintic treatments was only 34.7% when compared to the total number of the 757 treatments which would have been systematically administrated to all these horses as it was routinely done (at least twice per year) during the years before implementing the SAT program. On four out of the five farms, these reductions were much more marked and showed very low variations when comparing their results. A lower reduction of the treatment number of only 35% was revealed in horses on farm S. The reasons were unknown, but this farm—set up more recently—had quite a frequent turn-over of horses (and horse owners).

The results of the flotation/sedimentation technique will be presented in a different context.

There was no report of a clinical problem due to cyathostomin or even *Strongylus vulgaris* infection during the entire study period in any of the five farms.

It has to be mentioned that Farm C carried out a winter treatment once a year. This moxidectin based treatment was administrated to the 24 horses of Farm C between December and mid-January, with the aim to prevent any potential *S. vulgaris* infection. This treatment was administrated independently of the presented scheme and without any coprological analysis. The first annual monitoring of Farm C always occurred four to five months after the winter treatment.

## 4. Discussion

The presented data demonstrate that a continuous coprological analysis twice a year is easily feasible under conditions of riding farms with several horse owners. Undoubtedly, a major condition for a successful implementation of such a control scheme is the positive consent of the farm owner or manager. However, this field study clearly demonstrates that many horse owners—all owners in this study had to pay for the analyses—are prepared to pay for such a well-driven parasite control, despite some economic concern about the laboratory costs for fecal analyses [20]. Consequently, coproscopy-driven parasite control is reasonably adaptable and mostly successful, an additional advantage is that any kind of anthelmintic treatment can be made in a parasite-specific way according to the detected parasite spectrum. In some cases, a winter treatment is included in the parasite-control management, as seen here in Farm C. In our case, the influence of this treatment could not be evaluated but the influence of this treatment on the parasitic pressure deserves further attention.

The overall reduction of the number of anthelmintic treatments of 65.3% during the study period between autumn 2014 and spring 2020 in all these horses is relevant because before the beginning of the SAT-programme, all horses of this study have regularly been treated two or three times a year without any previous coprological analyses. The findings are of particular importance also with regard to AR development as they conform with Leathwick et al. [4] who found that for achieving a significant delay of AR development the number of annual treatments needs to fall to two or less per year. The fact that only one third needed a treatment prevented the other two thirds from overuse of anthelmintics which could result in increased resistance.

Such a reduction—comparable with figures previously achieved in more temporally limited studies [11,15]—is even more important when considering that the presently analyzed horses are of all age classes and from five different farms of France and Switzerland.

Also, the achieved overall reduction clearly shows the potential for a reduction of the number of anthelmintic treatments on horse riding farms when appropriately done [17]. These data allow to conclude that a selective anthelmintic treatment scheme can be successfully implemented on riding horse farms with varying numbers of horses or horse owners, or, under difficult management (e.g., frequent moving in/out of horses or horse owners) and pasture conditions.

Furthermore, such a continuous helminth monitoring program not only provides a key element for the successful management of cyathostome parasites but also forms the basis for any specific treatment and prevention of other helminth infections. With regards to the latter, it has to be underlined that there was not any single report of another parasite-associated disease problem on any of these farms for the entire study period. However, as also indicated by Tyden et al. [21], any monitoring procedure should regularly include a reliable detection method for *S. vulgaris* in order not to miss the re-emerging of this highly pathogenic helminth.

## 5. Conclusions

The strict administration of a selective anthelmintic treatment (SAT) program on all equids (90 horses, 3 ponies) of five riding horse farms in France and Switzerland over a period of six years reduced the total number of anthelmintic treatments by 65.3%. This reduction demonstrates the considerable potential of an SAT program for a long-term reduction in the number of anthelmintic treatments.

## Figures and Tables

**Table 1 animals-10-02395-t001:** This table represents the number of horses and the number of tests performed during the six-years period. The five farms are individually represented by the letters: C, N, B, S, and G.

Number of Horses and Number of Tests in Egg per Gram (EpG)	Farm C	Farm N	Farm B	Farm S	Farm G	Total
Number of horses tested per farm	22	7	15	29	20	93
Number of tests made during six years	168	76	122	214	177	757
Number of tests EpG ≥ 200 (absolute)	38	26	16	139	44	263
Number of tests EpG ≥ 200 (%)	22.6	34.2	13.1	64.9	24.8	34.7
Number of tests EpG ≤ 200 (%)	77.4	65.8	86.9	35.1	75.2	65.3

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
