# Peer review of "Preliminary Data from Six Years of Selective Anthelmintic Treatment on Five Horse Farms in France and Switzerland"

_animals, 2020, doi:10.3390/ani10122395_

Round 1

Reviewer 1 Report

The study is of interest, even the SAT system is not new and it has been applied in previous studies. Nevertheless, the interest of presented data is due to the long monitoring procedure and the diverse farms where the SAT was applied.

Overall, the manuscript is concise, the methods are correct and the conclusions are acceptable although the authors declare that there was no clinical problem due to cyathostomin or even S. vulgaris infections (lines 109 and 141). Nevertheless, it must be noted that in some farm a winter treatment with a moxidectin product was given and it should be better highlighted (or commented) by the authors.

The sensitivity of EPG should be clarified

Lines 57 and 99: clarified resp.

Figure 1 should be removed; as it is, do not add further information to that described in the manuscript; to note that the percent of reduction is reported three time in the figure.

Figure 2: the title is misleading; in the figure there is no comparison between strategic vs selective treatment, but the total number of fecal samples and the number of samples >200 EpG.

Author Response

Cover Letter. Answer to the reviewers

The authors are very grateful to the reviewers for their careful evaluation of our manuscript and for all the constructive inputs.

We have tried to answer all the various questions and to fulfill the requests having been put forward.

  1. Minor changes have been directly added in the article and are indicated by the “Word revision”. All comments made by the reviewers have been carefully considered.
  2. Major points” by one reviewer:

To begin with -  as the senior author and as a scientist with many years experiences in publishing in international peer-reviewed journals - I would like to emphasize with regard to the comment of the reviewer about a lack of robust controls that this is a preliminary retrospective analysis of field data generated over several years under “real” field conditions. Such a study has nothing to do with specifically designed field studies but reflects the real situation of anthelmintic drug administration under field conditions. I therefore would like to defend the present study as it stands by its concept because I am convinced that it undoubtedly reflects a practical approach and situation, in this case observed over a period of several years. Perhaps the reviewer has not realized that the control group is reflected by the same horse farms/horses in case they would have been regularly administering at least two anthelmintic treatments per year – as it also happened on these farms before switching to the selective approach.

Furthermore, I would like to mention that – not only for reasons of time and effort - a comparative study with farms where a strategic anthelmintic treatment is regularly done would not be possible because on these farms the horses are just regularly treated without any preceding coprological analyses.

Furthermore, under such given field conditions it is not possible – indeed never in live animals - to evaluate whether this strategy is effective at controlling the parasite burden as the coprological analysis mediates a reflection of the parasite egg excretion as a parameter for the contamination of the environment. Under no conditions this is a reflection of the parasite burden. But it is worldwide well established that the parasite egg excretion reflects a widely used measure for experimental studies, e. g. on the efficacy of anthelmintic drugs.

However, I fully agree with regard to the lack of a statistical analysis. We have seriously discussed this issue with an expert statistician who agreed that under natural conditions the variables do not allow do to any other test then a descriptive presentation of the data, as it actually is.

The reviewer has criticized that there is no attempt to evaluate the level of anthelmintic resistance on these farms. This has not been the objective of this study so there was no need to check for this in the given approach.

Regarding Introduction

Since the study/ms is designed as a Short Communication and because the issue of anthelmintic resistance of cyathostomes and also Parascaris equorum is a well-known and frequently described phenomenon, we feel that there is no need to extend this part further. We often see it even much shorter and all-inclusive in full papers addressing the problem of anthelmintic resistance. The treatment strategy is the same for each of these farms, but the administered drugs vary from farm to farm according to their veterinarian’s proposal. This has no immediate effect on the administration of the selective treatment scheme as this varies always from farm to farm.

As it is mentioned there is one treatment strategy which is obviously a targeted selective treatment of horses with a EpG-level > 200 EpG.

Every farm and horse owner administer the drug obtained by his veterinarian according to the indicated dosage.

Regarding Material and Methods

Due to the fact that it is a Short Comm and because the McMaster method is an internationally recognized standard method for quantitative coprological analyses we think that there is no need for a more detailed description of it.

Regarding Results

We fully agree with the reviewers that the results are preliminary.

Regarding Discussion

Due to the fact that it is a Short Comm., the discussion part has been kept short and it just and only reflects the obvious findings of the achieved results in a simple way, mainly in order not to exaggerate the interpretation of the results.

Reviewer 2 Report

the manuscript refers the results of a 6 year study in field about the effectiveness of selective anthelmintic treatment. I think that it is a cutting edge topic and it is worth of publishing in its present form

Claims are farmacoresistance of equine nematodes, this topic is very significant and claims are convincing. The discussion is well structured.

The manuscript stands out for its quality, it reports the results on a large scale study, dealing with the drug resistance to anthelmintic treatment in horses. This topic, mostly referred to strongyles is considered a major issue by EVPC, also and the Authors are authorities in the matter.

The results should be considered as preliminary and statistical analysis should be applied to further studies.

Author Response

(The authors gave the same response as above.)

Reviewer 3 Report

This study addresses the effect of managing anthelminthic treatment to horses in order to reduce the development of anthelmnthic resistance.

The authors should consider the following:

Major points

Although the authors have described  a reduction in treatment there is no statistical evaluation of this.  This is mainly due to the lack of robust controls.  For instance data should be compared to farms where such a strategy was not in place.  

There is no attempt to determine if this strategy is effective at controlling both parasite burden and prevalence of infection - again more controls either comparing to data acquired before the new strategy was introduced or by comparing with similar farms where the standard treatment regime was in place

Importantly there is no attempt at evaluating the level of AR on these farms.  Egg counts before and after each type of treatment would be required in both treatment regimes.

Introduction

This is rather brief and needs more detail -

Current status of anthelminthic resistance in horses and the potential problem this causes. 

What the current treatment strategies are, including pasture management, etc

Methods

These are inadequate -details of reagents and experimental procedure are required.

Details of treatment required - e.g. drugs used and dosage.

No description of statistical analysis

Results

Very poorly presented no attempt at statistical analysis

Discussion

Quite superficial - no substantive or statistically significant results

Author Response

(The authors gave the same response as above.)

Round 2

Reviewer 3 Report

This study is still significantly flawed, there is no statistical analysis or control groups - either in the from of anthelminthic treatment prior to the new treatment regime or with other horse farms not participating in the new treatment regime.

Author Response

We cannot follow the requests from this reviewer as they do not match the message explained in this manuscript. As explained there are no "control" groups because this manuscript is a description of a field analysis as it happens in real life (we did not ask these farms to follow the SAT-programme, but we choose them because the were doing the SAT-programme). We are just demonstrating a fact. 

In our opinion there is no need to make complicated stats to show something very simple. 

Concerning the English language, the article has been reviewed by a native english speaker.